# The epidemic characteristics of *Mycoplasma pneumoniae* infection among children in Anhui, China, 2015–2023

Bing Chen,[1] Ling-yu Gao,[2] Qiu-ju Chu,[3] Ting-dong Zhou,[1] Yang Tong,[1] Ning Han,[1] Ai-hua Wang,[1] Qiang Zhou[1]

**ABSTRACT** The number of pediatric respiratory tract infection cases in China has significantly increased this year, and *Mycoplasma pneumoniae* is one of the main pathogens. This study aimed to investigate the epidemiological characteristics of *M. pneumoniae* in children in the Anhui region and to provide evidence for the prevention and control strategies of *M. pneumoniae* in children in this region. A total of 66,488 pediatric patients with respiratory tract infection were enrolled from January 2015 to November 2023 in this study. The results of this study exhibited that *M. pneumoniae* infection in the Anhui region was characterized by a high positive rate during 2021–2023, especially this year is considered a year of pandemic for *M. pneumoniae* infection. Moreover, the positive rate of *M. pneumoniae* in female children is significantly higher than in male children, and the infection rate of *M. pneumoniae* in children increases significantly with age, particularly in school-aged children.

**IMPORTANCE** The number of pediatric respiratory tract infection cases in China has significantly increased this year, and *Mycoplasma pneumoniae* is one of the main pathogens. This study aimed to investigate the epidemiological characteristics of *M. pneumoniae* in children in the Anhui region and provide evidence for the prevention and control strategies of *M. pneumoniae* in children in this region.

**KEYWORDS** *Mycoplasma pneumoniae*, epidemic, children, COVID-19, respiratory tract infection

*M*ycoplasma pneumoniae is one of the leading pathogens that cause respiratory tract infections in humans, especially in children and adolescents. *M. pneumoniae* has become a common pathogen of pediatric community-acquired mycoplasma pneumonia (CAMP), accounting for 10%–40% of CAMP cases (1–3). In recent years, the incidence of *M. pneumoniae* pneumonia in children has increased significantly in the post COVID-19 epidemic era (4, 5). *M. pneumoniae* infection can cause respiratory diseases, such as acute and chronic respiratory infections, asthma, and bronchitis. In addition, *M. pneumoniae* may cause extrapulmonary symptoms, such as diarrhea, inflammation of the central nervous system, otitis media, arthritis, or acute myocardial injury (6). These extrapulmonary symptoms may be the result of excessive immune response or inflammation triggered by *M. pneumoniae* pathogen, which can eventually develop into a life-threatening disease (7).

*M. pneumoniae* is the smallest bacterium without a cell wall (8). Infection caused by droplet transmission during close contact can facilitate the spread of the pathogen from person to person, leading to an epidemic outbreak. *M. pneumoniae* infection is widely distributed worldwide, and its epidemic occurs periodically (9). Usually, regional epidemics occur every 3–7 years, and each epidemic may last for 1–2 years (10–12). Previous studies have shown that pandemics of *M. pneumoniae* occurred in Beijing, China, and the United Kingdom in 2011–2013 and 2015–2016, respectively.

Address correspondence to Qiang Zhou, zhouqiang1973@163.com.

Bing Chen, Ling-yu Gao, and Qiu-ju Chu contributed equally to this article. The author order was determined on the basis of seniority.

The authors declare no conflict of interest.

See the funding table on p. 9.

Unfortunately, there has been an increase in *M. pneumoniae* cases pneumonia (MPP) in China recently. Data revealed that *M. pneumoniae* infection accounted for approximately 70% of all cases of CAP in children aged 5 years and older. The epidemiological characteristics of MPP may vary due to factors, such as seasonality, geographic region, and genotype, and may also be related to population immunity and the maturity of the immune system (1, 8, 13, 14). However, there have been few studies focusing on the epidemiology and dynamic changes of MPP in a large number of pediatric patients in recent years.

To better understand the epidemiological characteristics of *M. pneumoniae* in children in China, this study retrospectively analyzed the children with respiratory tract infection who were admitted to the Second Affiliated Hospital of Anhui Medical University from 2015 to 2023 and collected the basic information and serological results of *M. pneumoniae* in the children. The distribution of infections was analyzed based on factors, such as sex, age, month, and pre- and post-epidemic conditions, which can provide evidence for the clinical diagnosis, treatment, and prevention and control strategies of *M. pneumoniae* in children in the local region.

## MATERIALS AND METHODS

### Study patients

A total of 66,488 pediatric patients with respiratory tract infection, including 37,182 boys and 29,306 girls, ranging in age from 1 month to 14 years, were enrolled in the Second Affiliated Hospital of Anhui Medical University from January 2015 to November 2023 in this study. The inclusion criteria were as follows: (1) the cases presented with clinical manifestations of upper respiratory tract infection, pneumonia and bronchitis, such as cough, nasal congestion and sore throat; (2) the data of the pediatric patients were complete, and the antibody titer results of *M. pneumoniae* were performed in the hospital. The exclusion criteria were as follows: (1) patients over 14 years old; (2) patients with trachea or lung dysplasia or other congenital diseases; (3) patients with heart, liver, kidney, and other basic diseases; (4) Patients with immune deficiency or long-term use of immunosuppressants; (5) people with a history of tuberculosis. The study was a retrospective study, all patient data were reported anonymously, and the study was conducted in accordance with the Declaration of Helsinki (revised in 2013) and approved by the Ethics Committee of the Second Affiliated Hospital of Anhui Medical University.

### Detection of *M. pneumoniae* antibody titers

Peripheral blood was extracted from the patient and centrifuged at 3,000 rpm($1,006 \times g$) for 10 min. Then, the total antibody titer of *M. pneumoniae* in human serum was measured by passive agglutination method using the *M. pneumoniae* antibody detection kit (Fuji Ruibai Co., Ltd., Japan). All operations were strictly performed according to the manufacturer's instructions. A serum *M. pneumoniae* antibody titer of ≥1:160 was considered as *M. pneumoniae* infection in the patient (10).

### Statistical analysis

Statistical analysis was performed by using Graphpad 8.0 (Graphpad Software, USA). The continuous variables were exhibited as mean ± standard deviation (M ± SD), and the categorical variables were exhibited as numbers (%). Two groups of data that met the normal distribution and homogeneity of variance were analyzed using Student's unpaired *t*-test, and the data that did not meet normal distribution and homogeneity of variance were analyzed using the Mann–Whitney U-test. Comparisons of multiple groups were performed using a one-way analysis of variance (ANOVA), followed by Tukey's honest significant difference test. Comparisons between categorical variables were performed using χ or Fisher exact test. Bonferroni correction was applied to address the problem of multiple comparisons and was calculated as each *P* value multiplied

by the number of comparisons. The differences were considered statistically significant when $P < 0.05$.

## RESULTS

### Positive rate of *M. pneumoniae* in different years

In this study, the total cases, positive cases, and positive rate of *M. pneumoniae* were calculated from 2015 to 2023. It was found that 66,488 children with respiratory tract infection were enrolled from 2015 to 2023, among which 21,331 cases were positive for *M. pneumoniae*, with a positive rate of 32.08%. The positive rate of *M. pneumoniae* showed significant differences among different years ($\chi^2 = 4772.8$, $P < 0.0001$). It can be observed from Fig. 1 that a peak epidemic of *M. pneumoniae* infection in children occurred in 2016 (20.06%), and a high prevalence of *M. pneumoniae* infection in children was observed in 2019–2023 (24.64%, 28.06%, 41.86%, 38.08%, and 47.89%) (see Fig. 1).

### Sex difference in positive rate of *M. pneumoniae*

Among the 21,331 positive cases of *M. pneumoniae*, 10,753 were male, and 10,578 were female, with a sex ratio of 1.017:1. The positive rate of *M. pneumoniae* in male pediatric patients was 28.92%, which was lower than that in female pediatric patients (36.09%), and the difference was statistically significant ($\chi^2 = 387.2$, $P < 0.0001$). In addition, the statistical analysis of the association between sex and the annual prevalence of *M. pneumoniae* was conducted using Pearson's χ test with Bonferroni correction (see Table 1). The results revealed that in 2016, the positive rates of *M. pneumoniae* in males and females were 18.64% and 21.89%, respectively, and there was no statistically significant difference between the two groups after Bonferroni correction (Bonferroni-corrected $P = 0.1116 > 0.05$). However, in other years (2015, 2017–2023), there were significant differences in the positive rates of *M. pneumoniae* between male and female children (Bonferroni-corrected $P < 0.05$).

### Changes in the positive rate of *M. pneumoniae* in different months

This study examined the differences in the positive rate of *M. pneumoniae* in different months and found no significant differences among different months ($P = 0.8$) (see Fig. 2A). In addition, we further analyzed the variation in the positive rate of *M. pneumoniae*

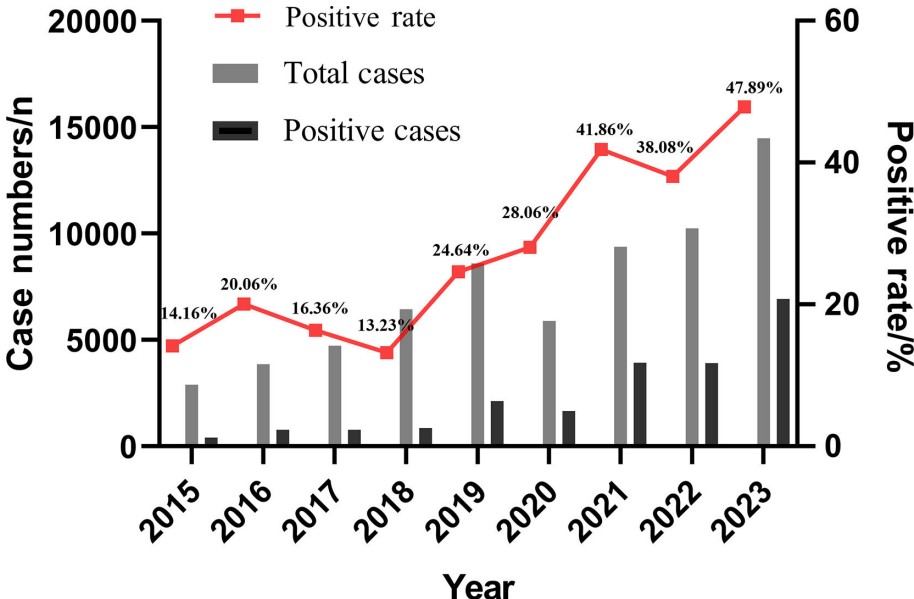

**FIG 1** Trend of positive rate of *M. pneumoniae* in different years.

**TABLE 1** Sex difference in the prevalence of *M. pneumoniae*

| Year | Male | | Female | | Uncorrected *P* value | Corrected *P* value[a] |
|---|---|---|---|---|---|---|
| | Positive cases (%) | Negative cases (%) | Positive cases (%) | Negative cases (%) | | |
| 2015 | 201 (12.52) | 1,404 (87.48) | 209 (16.19) | 1,082 (83.81) | 0.0049 | 0.0441 |
| 2016 | 405 (18.64) | 1,768 (81.36) | 369 (21.89) | 1,317 (78.11) | 0.0124 | 0.1116 |
| 2017 | 391 (14.77) | 2,257 (85.23) | 383 (18.38) | 1,701 (81.62) | 0.0009 | 0.0081 |
| 2018 | 427 (11.70) | 3,222 (88.30) | 425 (15.23) | 2,366 (84.77) | <0.0001 | <0.0001 |
| 2019 | 1,095 (22.58) | 3,754 (77.42) | 1,022 (27.31) | 2,720 (72.69) | <0.0001 | <0.0001 |
| 2020 | 814 (23.64) | 2,632 (76.38) | 837 (34.33) | 1,601 (65.67) | <0.0001 | <0.0001 |
| 2021 | 1,931 (37.54) | 3,213 (62.46) | 1,991 (47.11) | 2,235 (52.89) | <0.0001 | <0.0001 |
| 2022 | 1,948 (34.64) | 3,675 (65.36) | 1,953 (42.25) | 2,669 (57.75) | <0.0001 | <0.0001 |
| 2023 | 3,541 (44.01) | 4,504 (55.99) | 3389(52.74) | 3,037 (47.26) | <0.0001 | <0.0001 |

[a]Bonferroni-corrected *P* value was calculated as each *P* value multiplied by the number of tests ($n = 9$).

in different months of different years using a heatmap and found that the overall trend was that the positive rate of *M. pneumoniae* was generally higher from August to October, and that the positive rate of *M. pneumoniae* was significantly higher in 2021–2023 than in 2015–2019 ($\chi^2 = 22.38$, $P < 0.0001$) (see Fig. 2B). Furthermore, with the outbreak of the COVID-19 pandemic in January 2020, this study considered January 2020 as the boundary point to analyze the trend in the variation of the positive rate of *M. pneumoniae* before and after the COVID-19 pandemic. The results revealed that the positive rate of *M. pneumoniae* was generally higher after the COVID-19 pandemic compared with before (see Fig. 2C). Changes in the number of *M. pneumoniae* positive cases in different months can be found in Fig. S1.

## Difference of the positive rate of *M. pneumoniae* in different age groups

The positive rate of *M. pneumoniae* in children of different ages from 2015 to 2023 was calculated in this study. It was found that the positive rate of *M. pneumoniae* in infants and toddlers younger than 3 years old was 19.73% ± 11.09%, whereas the positive rate was 32.48% ± 12.02% in pre-schoolers aged between 3 and 6 years old. Moreover, the positive rate of *M. pneumoniae* was 39.27% ± 15.20% in the school-age group older than 6 years old. There was a significant difference in the positive rate of *M. pneumoniae* among these three groups (F = 480.0, $P < 0.0001$), as shown in Fig. 3. The difference of the number of positive cases of *M. pneumoniae* in different age groups can be found in Fig. S2.

## Comparison of *M. pneumoniae* antibody titers

The titer of serum *M. pneumoniae* antibody was detected by passive agglutination method in this study, and the proportion of different antibody titers in the positive cases was analyzed. It was found that the proportion of titers at 1:320 tended to stabilize, accounting for approximately 20%. During the period from 2015 to 2019, the proportion of antibody titers at 1:640 showed no remarkedly fluctuation. However, in the period from 2020 to 2023, with the increase in annual positive rate of *M. pneumoniae*, the proportion of titers 1:640 also increased. Conversely, the proportion of titers 1:160 showed the opposite trend (see Fig. 4A). Additionally, the correlation between the positive rate of antibody titers and the overall positive rate was analyzed, and the positive rates of the titer 1:160, 1:320, and 1:640 were all positively correlated with the overall positive rate, respectively (see Fig. 4B through D).

## DISCUSSION

The number of pediatric respiratory tract infection cases in China has significantly increased this year, and *M. pneumoniae* is one of the main pathogens. *M. pneumoniae* infection occurs worldwide and is a common pathogen for acute upper respiratory tract infections in children of all ages (15–17). Studies have shown there are significant

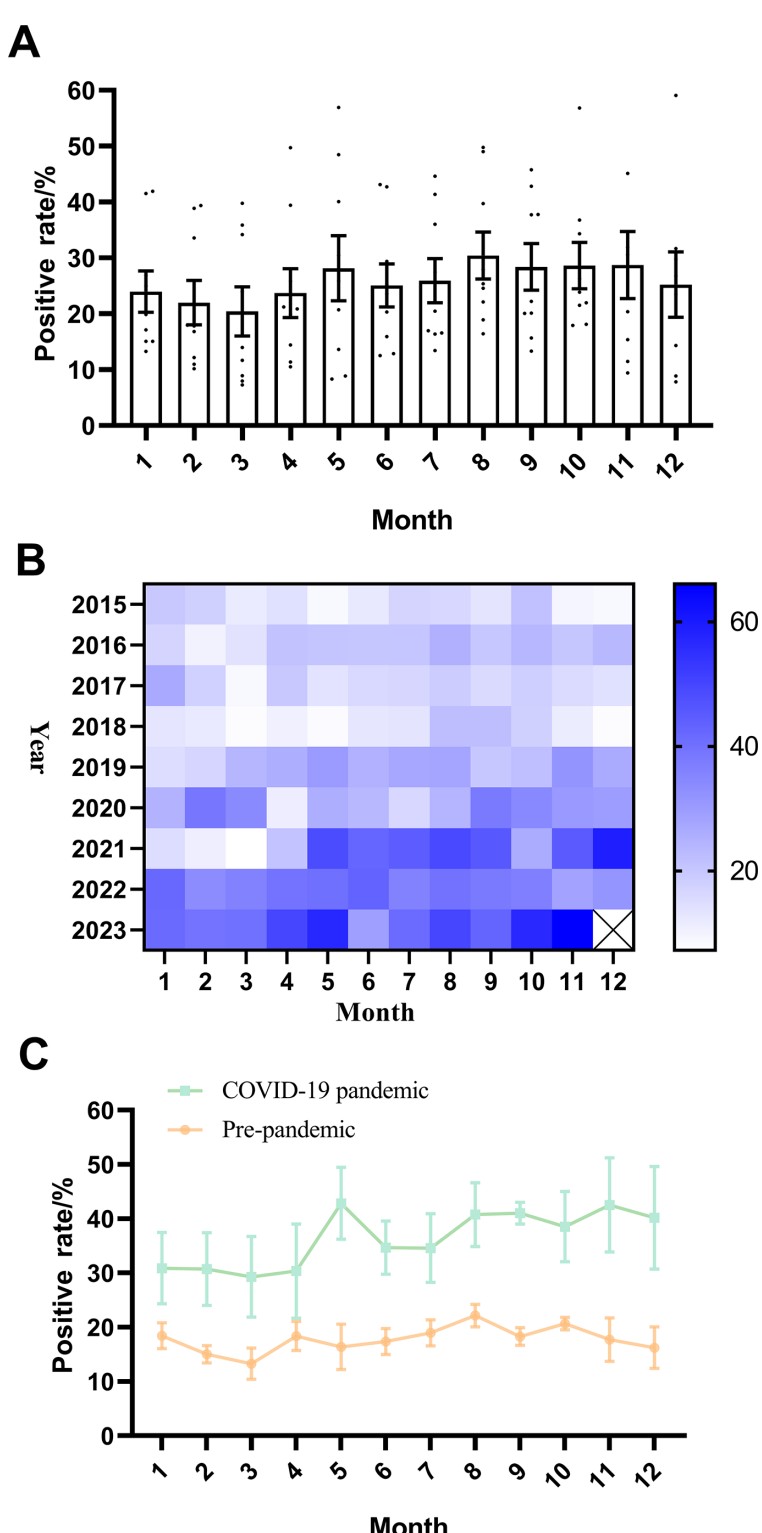

**FIG 2** Changes in the positive rate of *M. pneumoniae* in different months. (A) Changes in the positive rate of *M. pneumoniae* in different months. (B) The changes in the positive rate of *M. pneumoniae* in different months from 2015 to 2023 were demonstrated by heatmap. (C) Changes in the positive rate of *M. pneumoniae* in different months before and after the COVID-19 outbreak.

differences in the prevalence of *M. pneumoniae* infection among different countries and regions, different years and seasons, and different age groups (18). The prevalence of *M.*

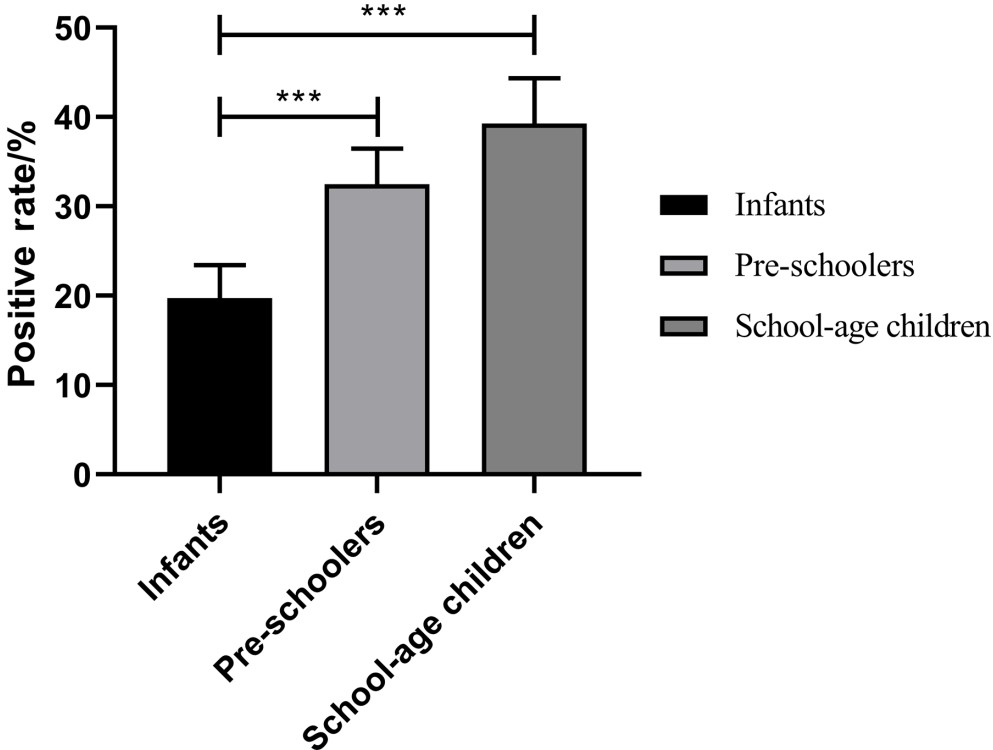

**FIG 3** Difference of the positive rate of *M. pneumoniae* in different age groups, ***:*P* < 0.001

*pneumoniae* infection varies greatly in different countries worldwide, ranging from 8.7% to 37.5% (19). Gao et al. exhibited that the positive rate of *M. pneumoniae* in children with respiratory symptoms in northern China was 37.5% (7). In contrast, Jiang et al. revealed that the positive rate of *M. pneumoniae* in children with respiratory symptoms in southern China was only 12.2% (20). In this study, the number of *M. pneumoniae* detection cases and positive cases from 2015 to 2023 in a tertiary hospital in Anhui region was analyzed. This study found that the positive rate of *M. pneumoniae* from 2015 to 2023 was 32.08%, which is between the above two(12.2% and 37.5%). During 2023, 14,471 cases of respiratory tract infections were tested, among which 6,930 cases were positive for *M. pneumoniae* infection, accounting for 47.89%, suggesting that respiratory tract diseases caused by *M. pneumoniae* infection played a major role in pediatric cases in 2023. In addition, *M. pneumoniae* infection has periodic epidemic characteristics. Eun et al. found that the peak of *M. pneumoniae* prevalence in South Korea occurs every 3–4 years (21). Studies have shown that the infection rate of *M. pneumoniae* in children in Xi'an, China was relatively low in 2018 and 2019 (25.82% and 24.13%) (22). This study found that the lowest positive rate of *M. pneumoniae* occurred in 2018, only 13.23% (852/5588), which was consistent with previous studies (22).

In addition, there are sex differences in the prevalence of *M. pneumoniae* infection. Previous studies have revealed that the prevalence of *M. pneumoniae* infection is higher in females than in males, suggesting that females may be more susceptible to *M. pneumoniae* (7, 23). Similarly, in this study, the positive rate of *M. pneumoniae* in male children was 28.92%, lower than the 36.09% in female children, and the difference was statistically significant, which was consistent with previous studies (22, 24, 25). However, the number of positive cases of *M. pneumoniae* infection in females is not always higher than in males during this time period. According to the data from the seventh national census in China (https://www.stats.gov.cn/sj/pcsj/), the ratio of male to female children in Anhui province is 117:100, which may result in the higher overall detection cases and positive cases of *M. pneumoniae* in males than those in females. Previous literatures (7, 8) also reported that the number of positive cases of *M. pneumoniae* in male children

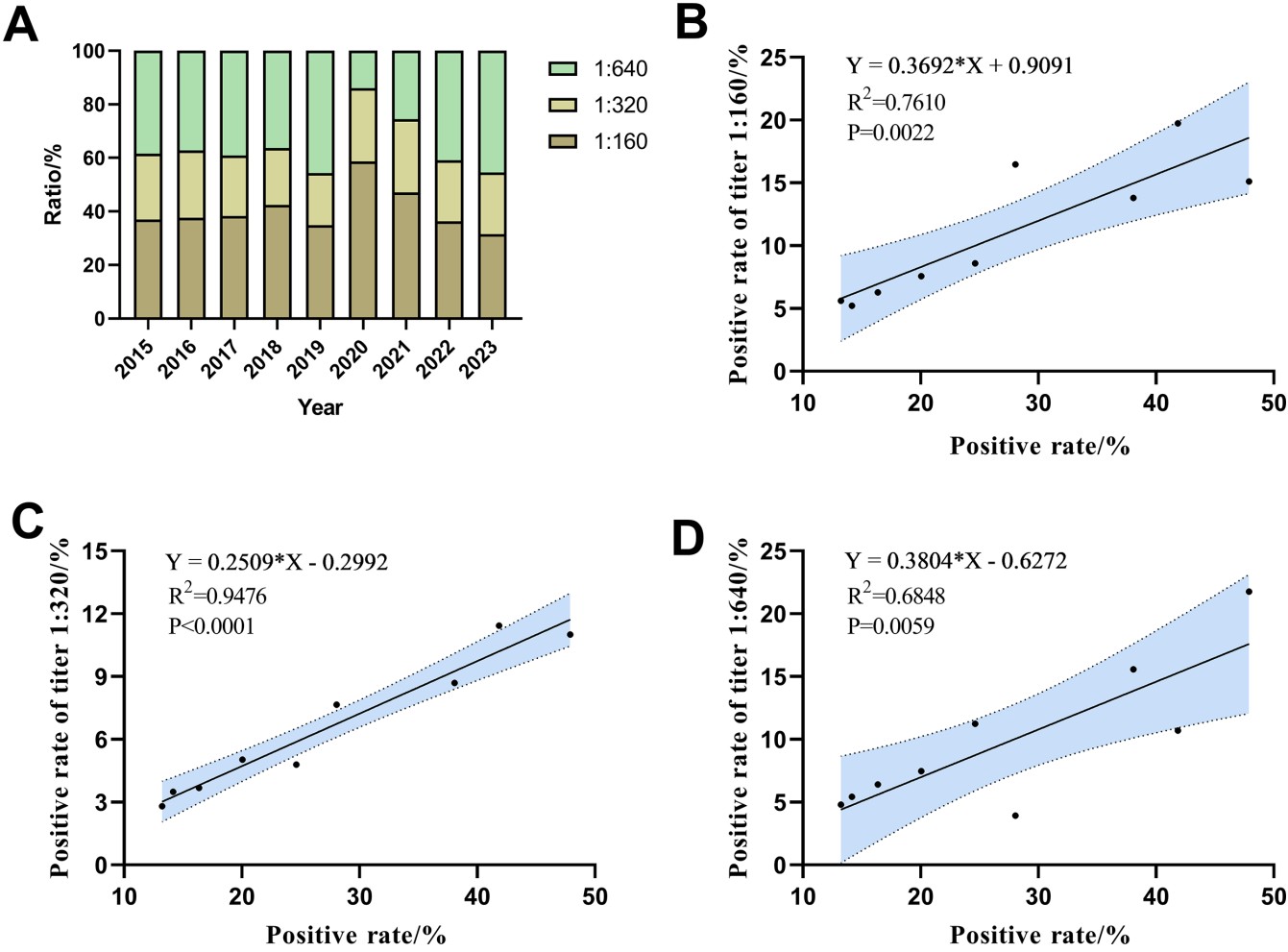

**FIG 4** Comparison of *M. pneumoniae* antibody titers. (A) Comparison of *M. pneumoniae* antibody titers between different years. (B) The correlation between the positive rate of titer 1:160 and the overall positive rate. (C) The correlation between the positive rate of titer 1:320 and the overall positive rate. (D) The correlation between the positive rate of titer 1:640 and the overall positive rate.

was higher than that in female children in other regions of China, whereas the positive rate for *M. pneumoniae* in female children was significantly higher than that in male children, which is similar to our findings. This may be related to differences in lifestyle between girls and boys. Girls tend to have indoor activities during breaks, and indoor environments are relatively enclosed, which is more conducive to the transmission of *M. pneumoniae* (24, 25). There are also studies showing that the prevalence of *M. pneumoniae* in girls is significantly higher than that of boys, which may be related to the difference in hormone levels of girls and boys, making girls susceptible to *M. pneumoniae*, and the specific mechanism is still unknown, and this sex difference needs to be further studied. Children with *M. pneumoniae* infections occur throughout the year, but the seasonal trends can annually vary (9). A study in northern China showed the peak epidemics of *M. pneumoniae* infections in autumn, whereas a study in southern China found the peak epidemics in summer (23). Additionally, research in Korea indicated that the peak of *M. pneumoniae* outbreaks occurs in autumn or winter. Other studies have shown that there are no significant seasonal fluctuations in the prevalence of *M. pneumoniae* infections (24). Similarly, this study conducted in Anhui province found no significant differences in *M. pneumoniae* infections between different seasons or months. Nevertheless, an overall trend observed was that the positive rate of *M. pneumoniae* was generally higher from August to October, possibly related to the increased temperature

during this period. In Anhui province, the temperature is higher, and people tend to gather in enclosed or semi-enclosed air-conditioned environments during August to October, which facilitates the transmission of *M. pneumoniae* (26, 27). Furthermore, *M. pneumoniae* outbreaks typically occur every 3–7 years and can last for 1–2 years (10–12). The most recent outbreaks were reported in 2013 and 2016 (25). The data from this study also showed that the peak epidemics for *M. pneumoniae* occurred in 2016 (20.06%). According to the epidemic pattern of *M. pneumoniae* outbreaks, the latest epidemic may last from autumn of 2019 to winter 2020 or spring 2021. Nevertheless, during the period from March 2020 to April 2021, the positive rate of *M. pneumoniae* did not increase significantly but significantly decreased. It was reported that the number of *M. pneumoniae* positive cases in Chengdu City decreased during the summer of 2020 (9, 10). Data in Finland and Japan also revealed a dramatic decrease in the incidence of *M. pneumoniae* in 2020 compared with 2012 and 2016 (28, 29). This may be due to the series of prevention and control measures issued by the National Health Commission of China after the outbreak of COVID-19 in 2020, including regular handwashing, restrictions on public gatherings, and wearing masks, effectively controlling the spread of the COVID-19 epidemic and markedly reducing the transmission of *M. pneumoniae* (4, 5, 9, 30). Notably, after April 2021, the positive rate of *M. pneumoniae* began to rebound, especially in children, with the highest positivity rate recorded for this year (47.89%). This may be because the COVID-19 pandemic delayed the timing of *M. pneumoniae* outbreaks.

*M. pneumoniae* infection can occur in any age group, especially in preschool and school-age children (31–33). The highest prevalence of *M. pneumoniae* is in the 7–10-year-age group, whereas in Australia, the highest prevalence of *M. pneumoniae* is in the 5–9-year-age group (34, 35). One study exhibited that the prevalence of *M. pneumoniae* in children older than 7 years old in South China is the highest. The results of this study showed that the positive rate of *M. pneumoniae* in infants and toddlers younger than 3 years old was 19.73% ± 11.09%, whereas in pre-schoolers aged between 3 and 6 years old, the positive rate was 32.48% ± 12.02%. Moreover, the positive rate of *M. pneumoniae* was 39.27±15.20% in the school-age group older than 6 years old, which is consistent with the results of most studies (9, 10). This indicates that the prevalence of *M. pneumoniae* infection in children increases significantly with age. School-age children between 6 and 14 years are more susceptible to infection, which may be due to their immature immune system and their frequent exposure to crowded places, such as kindergartens and schools, increasing the likelihood of respiratory pathogen infections and cross-transmission (23, 36). On the other hand, the prevalence of *M. pneumoniae* in infants aged 0–3 years is low, which may be related to the simple living environment and predominantly breastfeeding in this age group (22).

There are several limitations to this study. On the one hand, this study is a single-center retrospective analysis to explore the prevalence of *M. pneumoniae* infection in Anhui province based on clinical data from a large comprehensive tertiary hospital. To comprehensively analyze the epidemic patterns of *M. pneumoniae* in the region, multi-center analysis of the prevalence of *M. pneumoniae* infection can be considered in the future. On the other hand, the diagnosis of *M. pneumoniae* infection in this study was only based on the results of serum *M. pneumoniae* antibodies (passive agglutination method), and the antibody titer ≥1:160 was considered as recent *M. pneumoniae* infection.

In conclusion, the results of this study exhibited that *M. pneumoniae* infection in the Anhui region was characterized by a high positive rate during 2021–2023, especially this year is considered a year of pandemic for *M. pneumoniae* infection. Moreover, the positive rate of *M. pneumoniae* in female children is significantly higher than in male children, and the infection rate of *M. pneumoniae* in children increases significantly with age, particularly in school-aged (7–14 years old) children. Notably, during the period from April 2020 to March 2021, the positive rate of *M. pneumoniae* in children remained at a low level, indicating that the preventive and control measures for COVID-19 might have effectively controlled the transmission of *M. pneumoniae* infection.

## ACKNOWLEDGMENTS

This work was supported by the Foundation of Anhui Educational Committee (No. 2023AH053165).

B.C., Y.-L.G., Q.-J.C., and Q.Z. participated in study design, data analysis, and manuscript writing. T.-D.Z., Y.T., N.H., and A.-H.W. participated in data collection, data sorting, and manuscript modification. All authors approved the final version of the manuscript submitted.

## AUTHOR AFFILIATIONS

[1]Department of Clinical Laboratory, The Second Affiliated Hospital of Anhui Medical University, Anhui, China
[2]Institute of Dermatology, Chinese Academy of Medical Sciences and Peking Union Medical College, Nanjing, China
[3]Department of Clinical Laboratory, Hangzhou Xixi Hospital, Hangzhou, China

## AUTHOR ORCIDs

Bing Chen http://orcid.org/0000-0001-5986-7053

## FUNDING

| Funder | Grant(s) | Author(s) |
| --- | --- | --- |
| Foundation of Anhui Educational Committee | 2023AH053165 | Bing Chen |

## AUTHOR CONTRIBUTIONS

Bing Chen, Conceptualization, Data curation, Formal analysis, Funding acquisition, Methodology, Project administration, Writing – original draft, Writing – review and editing | Ling-yu Gao, Conceptualization, Data curation, Formal analysis, Methodology, Project administration, Supervision, Writing – original draft, Writing – review and editing | Qiu-ju Chu, Conceptualization, Formal analysis, Methodology, Supervision, Writing – original draft, Writing – review and editing | Ting-dong Zhou, Formal analysis, Investigation, Methodology, Validation | Yang Tong, Investigation, Methodology, Supervision | Ning Han, Formal analysis, Investigation, Methodology | Ai-hua Wang, Methodology, Supervision | Qiang Zhou, Conceptualization, Formal analysis, Investigation, Methodology, Supervision, Writing – review and editing

## DATA AVAILABILITY

The authors confirm that the data supporting the findings of this study are available within the article and supplemental material.

## ETHICS APPROVAL

This study was approved by the Institutional Ethics Committee of The Second Affiliated Hospital of Anhui Medical University (No. SZR2021038), and it was in compliance with national legislation and the Declaration of Helsinki guidelines.

## ADDITIONAL FILES

The following material is available online.

### Supplemental Material

**Supplemental figures (Spectrum00651-24-s0001.docx).** Fig. S1 and S2.

### Open Peer Review

**PEER REVIEW HISTORY (review-history.pdf).** An accounting of the reviewer comments and feedback.

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
