## [Reviewer comments · Microbiology Spectrum]

Microbiology Spectrum

The Epidemic characteristics of *Mycoplasma pneumoniae* infection among children in Anhui, China, 2015-2023

Bing Chen, Lingyu Gao, Qiu-ju Chu, Ting-dong Zhou, Yang Tong, Ning Han, Ai-hua Wang, and Qiang Zhou

Corresponding Author(s): Qiang Zhou, The Second Affiliated Hospital of Anhui Medical University

Review Timeline:

Submission Date:	March 11, 2024
Editorial Decision:	May 2, 2024
Revision Received:	May 20, 2024
Editorial Decision:	June 7, 2024
Revision Received:	June 19, 2024
Accepted:	July 17, 2024

Editor: Deena Altman

Reviewer(s): Disclosure of reviewer identity is with reference to reviewer comments included in decision letter(s). The following individuals involved in review of your submission have agreed to reveal their identity: Robert Doug Hardy (Reviewer #1)

Transaction Report:

DOI: <https://doi.org/10.1128/spectrum.00651-24>

Re: Spectrum00651-24 (The Epidemic characteristics of Mycoplasma pneumoniae infection among children in Anhui, China, 2015-2023)

Dear Dr. Bing Chen:

Thank you for the privilege of reviewing your work. Below you will find my comments, instructions from the Spectrum editorial office, and the reviewer comments.

Revision Guidelines

Sincerely,
Deena Altman
Editor
Microbiology Spectrum

Reviewer #1 (Comments for the Author):

The subject matter of the manuscript is interesting and worthwhile to explore.

Two major points to address:

1. Lines 304-307 - Can this assumption be validated by existing literature. If this assumption cannot be rigorously validated, then the papers conclusions cannot be accepted as meaningful.

2. The papers analysis seems to be principally based on the PERCENTAGE of subjects with M. pneumoniae infection. While the PERCENTAGE of subjects with M. pneumoniae infection might be of use for some analysis, the ACTUAL NUMBER of M. pneumoniae cases per time period (year, month), age, gender is the correct measurement to compare. In fact, the PERCENTAGE and the ACTUAL NUMBER of M. pneumoniae cases are not parallel for many of the stated outcomes. In addition, the PERCENTAGE of M. pneumoniae cases is dependent on the annual number of other respiratory infections (RSV, adenovirus, influenza, parainfluenza, human pneumometavirus, S. pneumonia, C. pneumoniae, and other) as well as immunization to theses other infections.

Reviewer #2 (Comments for the Author):

The authors reported a study on the epidemic characters of M. pneumoniae infections between 2015 and 2023 in children in Anhui, China and found an increased positive rate during 2021-2023, as well as differences between gender and ages. This study added timely local information to the current M. pneumoniae outbreaks which helped us to better understand the epidemic characters of this outbreak in Anhui, China. There are some questions /comments for the authors:

1. Line 120-121: Please provide the approved document number from the Ethics Committee of the institute.

2. Line 123: Please provide the g-force used for centrifuging the blood sample.

3. For statistical analysis, when applying the t-test, was there a test performed to assure the data were normally distributed? A paired sample t test was used for comparing the positive rates between male and female patients in different years. This may not be appropriate since paired sample t test can only compare the means for two related (paired) units. The male and female patients in this study were not related. Also post hoc test such as Tukey's honestly significant difference (HSD) is suggested in addition to the t test.

4. Results. Section 3.3 may be combined with 3.1 since both were about the analysis of M. pneumoniae positive rate in different times. For the comparisons of differences among multiple time points, only an overall p value was provided. A correction test, such as Bonferroni test should be performed. This is the same case for comparison of the genders in different years. It is also suggested stating the number of positive rate in the paragraph, especially for the point of interests, such as the year of 2016, 2019-2023. The positive rate during COVID should also be stated.

5. This study showed the data about the anti-M. pneumoniae antibody titers over years, which is important to understand the changes in population immunity. Are there any data on patients not infected with M. pneumoniae that could be used as "negative" control?

Reviewer #3 (Comments for the Author):

Although the immunological analysis between 2015 and 2023 is very interesting, but there several main limitations.

1, The study just investigate upper respiratory tract infection (URTI) in only one hospital, could not represent the situation of Anhui province or Mp infection which could include LRTIs.

2, only passive agglutination was used to detect Mp, which might underestimate the incidence. PCR should be included.

3, no ethical number had been provided.

Response to Reviewers

Dear Reviewers,

Thank you for your review of our manuscript “*The Epidemic characteristics of Mycoplasma pneumoniae infection among children in Anhui, China, 2015-2023*”. We appreciate the concerns and suggestions provided by the reviewers, and have revised our manuscript accordingly. At this time, we have re-submitted the revised manuscript through the Author Center, and we hope to have an opportunity to publish this paper in “*Microbiology Spectrum*”.

Below, please find our point-by-point responses to the comments of reviewers.

Comments from the reviewer:

Reviewer: 1

1. Lines 304-307 - Can this assumption be validated by existing literature. If this assumption cannot be rigorously validated, then the papers conclusions cannot be accepted as meaningful.

Response: Thanks for raising this important question. The results of this study **were compared with existing literatures**, and it was found that the epidemiological characteristics of *M. pneumoniae* infection in this study **were basically consistent with those reported in previous literatures**. For example, Studies have shown that the infection rate of *M. pneumoniae* in children in Xi'an, China was relatively low in 2018 and 2019 (PMID: 35685083). This study found that in 2018, the positive rate of *M. pneumoniae* in Anhui was the lowest, only 13.23% (852/5588). Furthermore, previous studies have revealed that the prevalence of *M. pneumoniae* infection is higher in females than in males, suggesting that females may be more susceptible to *M. pneumoniae* (PMID: 21822990, 36388772, 35573799). Similarly, in this study, the positive rate of *M. pneumoniae* in male children was 28.92%, lower than the 36.09% in female children, and the difference was statistically significant. In addition, this study revealed that the positive rate of *M. pneumoniae* decreased significantly in the early stage of the COVID-19 epidemic (from March 2020 to April 2021), which was consistent with previous studies (PMID: 36649824, 35551702, 34012847, 33910509, 35582916).

2. The papers analysis seems to be principally based on the PERCENTAGE of subjects with *M. pneumoniae* infection. While the PERCENTAGE of subjects with *M. pneumoniae* infection might be of use for some analysis, the ACTUAL NUMBER of *M. pneumoniae* cases per time period (year,

month), age, gender is the correct measurement to compare. In fact, the PERCENTAGE and the ACTUAL NUMBER of *M. pneumoniae* cases are not parallel for many of the stated outcomes. In addition, the PERCENTAGE of *M. pneumoniae* cases is dependent on the annual number of other respiratory infections (RSV, adenovirus, influenza, parainfluenza, human pneumometavirus, *S. pneumoniae*, *C. pneumoniae*, and other) as well as immunization to these other infections.

Response: Thanks for raising this important question. We carefully considered the issue you raised and **fully agree with your point of view**. Indeed, the increase in the detection of *M. pneumoniae* may be attributed to the prevalence of other respiratory pathogens (RSV, adenovirus, influenza, parainfluenza, human pneumometavirus, *S. pneumoniae*, *C. pneumoniae*, and other), leading to a decrease in the positive rate of *M. pneumoniae* infection. Therefore, we **have also compared** the number of positive cases of *M. pneumoniae* infection in different time periods, and genders according to your suggestions, and the results are **shown in the supplementary materials**. It was found that the positive rate of *M. pneumoniae* in this study is generally consistent with the trend of positive cases.

Reviewer: 2

1. Line 120-121: Please provide the approved document number from the Ethics Committee of the institute.

Response: Thank you for your suggestions. We **have added** the ethical number in the manuscript.

2. Line 123: Please provide the g-force used for centrifuging the blood sample.

Response: Thank you for your suggestions. We **have added** the size of centrifugal forces in the manuscript.

3. For statistical analysis, when applying the t-test, was there a test performed to assure the data were normally distributed? A paired sample t test was used for comparing the positive rates between male and female patients in different years. This may not be appropriate since paired sample t test can only compare the means for two related (paired) units. The male and female patients in this study were not related. Also post hoc test such as Tukey's honestly significant difference (HSD) is

suggested in addition to the t test.

Response: Thank you for your suggestions. In this study, normal distribution and homogeneity of variance were **carried out before t test analysis**, and two groups of data that met the normal distribution and homogeneity of variance were analyzed using Student's unpaired t-test and data that unmet normal distribution and homogeneity of variance were analyzed using the Mann-Whitney U-test. In addition, we **have checked and modified** the description of the statistical analysis and the results. Comparisons of multiple groups were performed using a one-way analysis of variance (ANOVA), followed by Tukey's honest significant difference post-hoc test for correction of multiple comparisons.

4. Results. Section 3.3 may be combined with 3.1 since both were about the analysis of *M. pneumoniae* positive rate in different times. For the comparisons of differences among multiple time points, only an overall p value was provided. A correction test, such as Bonferroni test should be performed. This is the same case for comparison of the genders in different years. It is also suggested stating the number of positive rates in the paragraph, especially for the point of interests, such as the year of 2016, 2019-2023. The positive rate during COVID should also be stated.

Response: Thank you for your suggestions. We **have added** the number of positive rates in the paragraph. we greatly appreciate your guidance on the statistical methods used in this research, and we **have made modifications** to the content in the manuscript based on your suggestions. We **have added** the Bonferroni correlation in Table 1 with reference to previous literature (PMID: 24736530).

5. This study showed the data about the anti-*M. pneumoniae* antibody titers over years, which is important to understand the changes in population immunity. Are there any data on patients not infected with *M. pneumoniae* that could be used as "negative" control?

Response: Thank you very much for your valuable questions. This study only included pediatric patients with respiratory tract infection, and did not conduct *M. pneumoniae* antibody titer testing in healthy populations or patients not infected with *M. pneumoniae*. However, previous studies have shown that the detection of *M. pneumoniae* antibody in healthy populations or patients with non-*M. pneumoniae* infection by passive agglutination method is usually negative or titers of 1:40 or 1:80. Therefore, in various expert consensuses on laboratory diagnostics of *M. pneumoniae* infection in

children in China, when the particle agglutination method is used to detect *M. pneumoniae* antibody, a serum antibody titer $\geq 1:160$ has been used as the reference standard for *M. pneumoniae* recent infection or acute infection(PMID: 32392951, 32392951, 37029345).

Reviewer: 3

1. The study just investigate upper respiratory tract infection (URTI) in only one hospital, could not represent the situation of Anhui province or Mp infection which could include LRTIs.

Response: Thank you very much for your comments. The problem you pointed out is also one of the limitations of this study, which we have elaborated in the discussion section. This study only retrospectively analyzed the epidemic characteristics of *M. pneumoniae* infection in pediatric patients with respiratory infections at our hospital from 2015 to 2023, which belongs to a single-center retrospective analysis. In future work, it can be further combined with multiple medical centers in Anhui province for comprehensive analysis. Besides *M. pneumoniae* infection, more pathogens of respiratory tract infection should also be considered, such as influenza, adenovirus, and so on.

2. only passive agglutination was used to detect MP, which might underestimate the incidence. PCR should be included.

Response: Thank you very much for your comments. It is undeniable that PCR testing for *M. pneumoniae* infection has high sensitivity and specificity, and has been widely used in clinical practice. However, in some small hospitals, especially in resource-limited countries and regions, PCR testing may not be available. Passive particle agglutination and immunochromatography for detecting serum antibodies against *M. pneumoniae* are still the main laboratory methods for clinical diagnosis of *M. pneumoniae* infection (PMID: 23076954, 37478238). Our hospital used to mainly rely on passive particle agglutination method to detect *M. pneumoniae* antibody titers to determine whether patients were infected with *M. pneumoniae*, without conducting PCR testing for *M. pneumoniae* nucleic acid. Therefore, this study did not include the results of PCR testing for *M. pneumoniae*. However, our hospital has now started to conduct PCR testing for *M. pneumoniae* infection, and the nucleic acid test combined with antibody detection is more reliable in clinical diagnosis of *M. pneumoniae* infection.

3. no ethical number had been provided.

Response: Thank you for your suggestions. We **have added** the ethical number in the manuscript.

We hope you and the reviewers will agree that we have addressed all the concerns raised by the reviewers in their previous review of the manuscript and that the integration of the reviewers' comments into the revised manuscript has substantially improved the manuscript. Thank you for your re-consideration of our manuscript and we are looking forward to your favorable decision.

Yours sincerely,

Dr. Zhou

E-mail: zhouqiang1973@163.com

Re: Spectrum00651-24R1 (The Epidemic characteristics of Mycoplasma pneumoniae infection among children in Anhui, China, 2015-2023)

Dear Dr. Qiang Zhou:

Thank you for the privilege of reviewing your work. Below you will find my comments, instructions from the Spectrum editorial office, and the reviewer comments.

Revision Guidelines

Sincerely,
Deena Altman
Editor
Microbiology Spectrum

Reviewer #1 (Comments for the Author):

Thank you for your responses.

In regard to analysis based on the PERCENTAGE of subjects with M. pneumoniae infection compared with analysis based on the ACTUAL NUMBER of M. pneumoniae cases, substantial statistical differences exist for age and gender. For age, the data is

clearly different statistically and much less dramatic. For gender, the data shows that in 9 years the percentage positive is greater for females. However, in terms of comparing total numbers positive, females are greater than males in only 4 of 9 years.

The comparisons in this manuscript for PERCENTAGE vs ACTUAL NUMBER of M. pneumoniae cases are not parallel for age and gender.

The PERCENTAGE of M. pneumoniae cases in the total number of children with respiratory tract infections is dependent on the number of other respiratory infections (RSV, adenovirus, influenza, parainfluenza, human pneumometavirus, S. pneumonia, C. pneumoniae, and other) as well as immunization to these other infections.

Conclusions should be adjusted to a full analysis of results.

Reviewer #2 (Comments for the Author):

There are significant improvements in the revised version. This reviewer's comments have been addressed.

Response to Reviewers

Dear Reviewers,

Thank you for your review of our manuscript “*The Epidemic characteristics of Mycoplasma pneumoniae infection among children in Anhui, China, 2015-2023*”. We appreciate the concerns and suggestions provided by the reviewers, and have revised our manuscript accordingly. At this time, we have re-submitted the revised manuscript through the Author Center, and we hope to have an opportunity to publish this paper in “*Microbiology Spectrum*”.

Below, please find our point-by-point responses to the comments of reviewers.

Comments from the reviewer:

Reviewer: 1

1. In regard to analysis based on the PERCENTAGE of subjects with *M. pneumoniae* infection compared with analysis based on the ACTUAL NUMBER of *M. pneumoniae* cases, substantial statistical differences exist for age and gender. For age, the data is clearly different statistically and much less dramatic. For gender, the data shows that in 9 years the percentage positive is greater for females. However, in terms of comparing total numbers positive, females are greater than males in only 4 of 9 years. The comparisons in this manuscript for PERCENTAGE vs ACTUAL NUMBER of *M. pneumoniae* cases are not parallel for age and gender. The PERCENTAGE of *M. pneumoniae* cases in the total number of children with respiratory tract infections is dependent on the number of other respiratory infections (RSV, adenovirus, influenza, parainfluenza, human pneumotmetavirus, *S. pneumoniae*, *C. pneumoniae*, and other) as well as immunization to these other infections.

Conclusions should be adjusted to a full analysis of results.

Response: Thank you very much for your careful review, and we agree with your points.

However, the comparisons in this manuscript for PERCENTAGE vs ACTUAL NUMBER of *M. pneumoniae* cases are not parallel for gender. We would like to further explain this discrepancy. According to the data from the seventh national census in China(<https://www.stats.gov.cn/sj/pcsj/>), the ratio of male to female children in Anhui province is 117:100, which may result in the overall detection cases and positive cases of *M. pneumoniae* in males is higher than that in females. Additionally, we compared the results of *M. pneumoniae* infections in male and female children in other regions of China as reported in previous literatures (PMID: 30016939; PMID: 31364532), and the results also showed the number of positive cases of *M. pneumoniae* in male children was higher

than that in female children, while the **positive rate for *M. pneumoniae* in female children was significantly higher than that in male children**. Therefore, the conclusion of this manuscript reveals the **positive rate** of *M. pneumoniae* in female children is significantly higher than in male children. In addition, the results of this study showed that the positive rate of *M. pneumoniae* in infants and toddlers under 3 years old was 19.73±11.09%, while in pre-schoolers aged between 3 and 6 years old, the positive rate was 32.48±12.02%. The positive rate of *M. pneumoniae* was 39.27±15.20% in the school-age group above 6 years old, which is consistent with the results of most studies (PMID: 34012847, PMID: 33910509). This indicates that the prevalence of *M. pneumoniae* infection in children increases significantly with age, particularly in school-aged children.

The above is our explanation of your comments. Furthermore, we would like to thank you once again for your very valuable comments on the manuscript, and we will be more careful in the future data processing and scientific research.

Characteristic	Total	M. pneumoniae cases	P value
Age, median (IQR)	16 (53)	11 (19)	
Children	3983 (50.8)	786 (19.7)	< 0.001
Adults	3852 (49.2)	341 (8.9)	
Gender			0.003
Female	3514 (44.9)	552 (15.7)	
Male	4321 (55.1)	575 (13.3)	

(PMID: 30016939)

	Pneumonia (n)	MPP (n, %)	χ^2	P
Sex			322.5	<0.0001
Male	16 346	5415 (33.1)		
Female	11 152	4885 (43.8)		

(PMID: 31364532)

We hope you and the reviewers will agree that we have addressed all the concerns raised by the reviewers in their previous review of the manuscript and that the integration of the reviewers' comments into the revised manuscript has substantially improved the manuscript. Thank you for your re-consideration of our manuscript and we are looking forward to your favorable decision.

Yours sincerely,

Dr. Zhou

E-mail: zhouqiang1973@163.com

Re: Spectrum00651-24R2 (The Epidemic characteristics of Mycoplasma pneumoniae infection among children in Anhui, China, 2015-2023)

Dear Dr. Qiang Zhou:

Accepted on the condition that there is one more comment from reviewer to be added:

"Thank you for your explanation. Consider adding your explanation to the manuscript discussion, such as: "the ratio of male to female children in Anhui province is 117:100, which may result in". This may clarify the seemingly discordant results to readers.

I would have the authors' let the readers know that "the ratio of male to female children in Anhui province is 117:100". This may clarify the seemingly discordant results to readers from populations with ratios of male to female children closer to 1:1."

Your manuscript has been accepted, and I am forwarding it to the ASM production staff for publication. Your paper will first be checked to make sure all elements meet the technical requirements. ASM staff will contact you if anything needs to be revised before copyediting and production can begin. Otherwise, you will be notified when your proofs are ready to be viewed.

Sincerely,
Deena Altman
Editor
Microbiology Spectrum

Reviewer #1 (Comments for the Author):

Thank you for your explanation. Consider adding your explanation to the manuscript discussion, such as: "the ratio of male to female children in Anhui province is 117:100, which may result in". This may clarify the seemingly discordant results to readers.